# On Compositional Generalization in Language Models

## Abstract

Recent language models appear to solve complex tasks that require logical reasoning with its rationale through a large number of parameters and instruction tuning. While step-by-step explanations have been introduced to improve the accuracy of the final prediction of the language models, there is still a lack of research on the reliability of the rationale. Therefore, the paper includes a study on the compositional reasoning ability of language models, as well as an analysis of the logical proof generated by them. By employing clear and straightforward semantics and syntax of a boolean expression, we observed and analyzed how language models generalize and solve the boolean formula. We classified boolean expressions based on their depth and empirically observed that language models not only struggle to comprehend more complex boolean expressions but also that their rationale is unreliable in affirming that the language models truly understand and solve the problem. From the perspective of understanding the structure of boolean algebra expressions, we discovered that language models inherently fail to generalize the compositional structure and often fail not only in calculating formulas but in grasping the input structure itself.

## 1 Introduction

The principle of compositionality, which states that complex structures are composed of simpler expressions, enables the generation of an infinite number of sentence expressions (Frege, 1884/1950; Chomsky, 1957; Montague, 1970). Besides sentence generation, this principle is a core tenet of reasoning. For example, dynamic programming or divide and conquer algorithm divides the entire problem into subproblems and finds the optimal solution using the intermediate solution of the subproblems (Cormen et al., 2001). In real world applications, such as research, law, or task planning, humans often encounter complex tasks that necessitate a step-by-step, compositional approach for them to be solvable. As human language is constructed systematically and humans justify various kinds of problems in a step-by-step way, compositional generalization is considered as important indicator for human-like intelligence (Simon, 1962).

Recently, large-scale language models have been applied for problems that require logical explanations with the help of emerging abilities like instruction understanding, reasoning and rationale generation (Wei et al., 2023; Kojima et al., 2023). There are several research directions studying how language models comprehend language and reasoning compositionally like humans (Lake & Baroni, 2018; Kim & Linzen, 2020; Dziri et al., 2023). Using these abilities, the explicit representation of step-by-step problem solving provides a mechanism of explanation and results in increases the final prediction accuracy. From this point of view, someone might expect that with this kind of rationale generation, the language models apply this step-by-step reasoning to a more complex problem and solve it.

Several datasets and studies have used natural language as an interface to formal problem solving with compositional reasoning, for example, datasets such as GSM8k (Cobbe et al., 2021) or SVAMP (Patel et al., 2021) use language as interface to solve elementary algebra problems. However, in these contexts, the problem relies on linguistic context for its meaning and is heavily influenced by pre-trained knowledge captured in large language models. Moreover, there are countless ways to arrive at a rationale, and the boundaries for steps, i.e., composition, are often unclear. Additionally, some versions of these reasoning tasks, such as elementary algebra require computaiton make it difficult

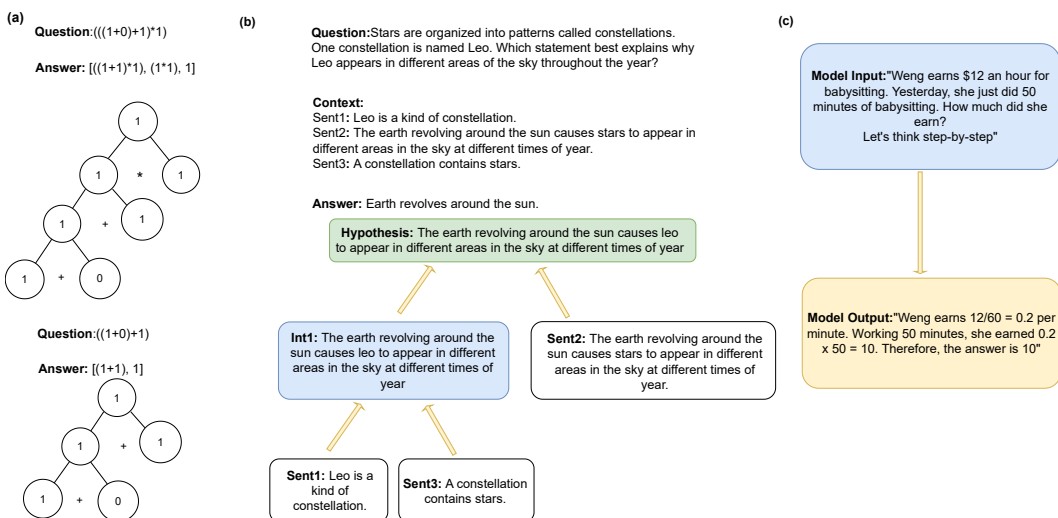

Figure 1: Comparison between boolean logic ((a)) and natural language reasoning Entailment Bank (Dalvi et al., 2022) ((b)) and Chain-of-Thought prompt (Wei et al., 2023) ((c)). The natural language sentences are simplified into variables of 0 and 1, allowing for a clear representation of logical computations between them. In (a), we can see the explicit compositional feature of a boolean expression. The boolean expression and its proof are composed of smaller compositions of the expression and its proof, respectively.

to discern whether the limitations in performance stems from calculations or the models ability to understanding and generate compositional structures. Unlike natural language reasoning datasets, which are heavily influenced by contextual meaning and plausibility, boolean algebra expressions are fundamentally and formally compositional (Wittgenstein, 1922) .Any proof can be expressed in a clear and step-by-step compositional manner. In contrast to other compositional reasoning datasets that rely on arithmetic calculations, the atomic operations in boolean algebra are simple, with a limited number of cases, as operands are binary, meaning that atomic operations can be expressed as two- or four-entry truth tables. By simulating structure with boolean algebra, we can demonstrate whether the transformer architecture can encode text in a purely formal compositional way and evaluate its ability to solve boolean expressions systematically, distinguishing boolean algebra from other existing datasets.

Our aims are to ascertain whether a language model can employ basic elements to construct a logical framework, prior to engaging in intricate logical, algebraic calculations and the relationships within complex natural language sentences (Figure 1). Additionally, we investigate the attributes of these logical constructions, particularly the proofs generated by the language model. The contribution of this paper is that

1. Using formal logic language datasets, we demonstrate that the transformer architecture, which are used for large language models, do not reliably encode compositional reasoning. Our results extend the claims of (Clark et al., 2020; Saparov & He, 2023; Dziri et al., 2023) that transformers fail to generalize to more complex reasoning tasks. We contribute findings that language models show difficulties in reasoning even in formal data where complexity is explicitly presented. This demonstrates that language models struggle to understand and generate compositional structure, which implies that the recent achievements in reasoning are not a result of language models systematical and structural understanding of tasks.

2. We present findings examining the failure points of the model's proofs. Our results show that when language models solve boolean expressions, the errors do not occur from computing components or propagation, but occur before calculating components in task representation. Transformers have difficulty in encoding boolean expressions that have more complex structures than those in the training data. This prevents the models from forming a generalizable representation of compositional inputs.

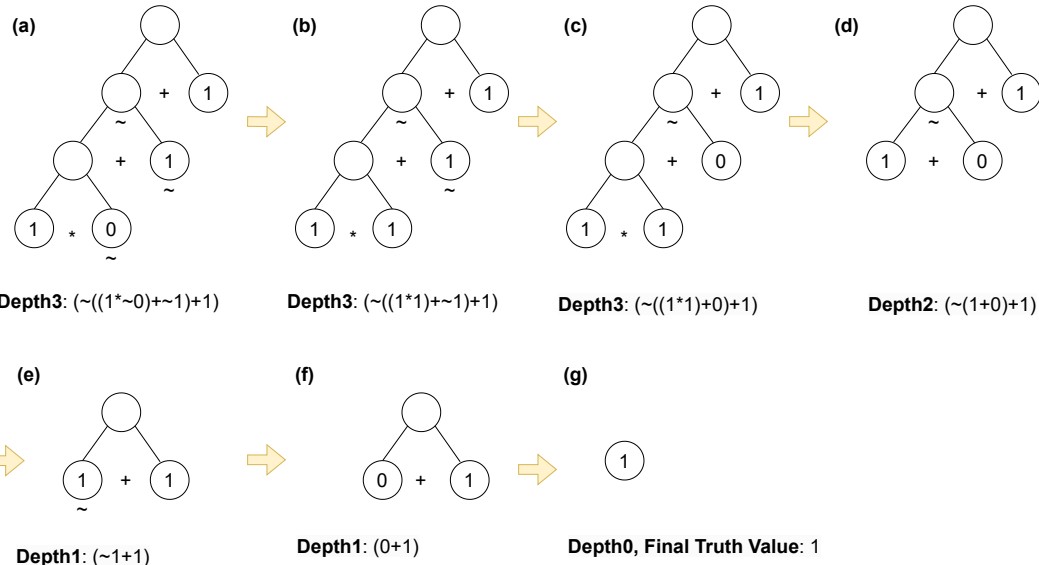

Figure 2: The proof step of boolean algebra expression. The model should generate the whole solution of boolean algebra calculation given input. For example, if the input is $(\sim ((1* \sim 0)+ \sim 1)+1)$, the model should generate the sequence $(\sim ((1*1)+ \sim 1)+1), (\sim ((1*1)+0)+1), (\sim (1+0)+1), (\sim 1+1), (0+1), 1$. (See Appendix B)

## 2 RELATED WORK

**Reasoning Ability of Language Model** By exploiting emergent reasoning and the self-justification ability of large-scale language models, task requiring complex reasoning appear to be solved with Chain-of-Thought prompting (Wei et al., 2023; Kojima et al., 2023) or selection-inference (Creswell et al., 2022). Due to the significant impact of generating the rationale on final prediction accuracy, there has been extensive research focused on creating and evaluating accurate rationales (Golovneva et al., 2023; Zhou et al., 2023; Wang et al., 2023). One question raised in our paper is whether the generated explanation of the transformer in the reasoning task is generated in a systematic way. Saparov & He (2023) shares similarity with our work, measuring the reasoning ability of language models with the ProntoQA dataset, which is expressed with predicate logic and containing false or fictional ontology, to explore the pure reasoning ability of language models. However, while Saparov & He (2023) focused on whether the language model can reason with predicates and quantifiers in natural language, we focus on whether the language model can grasp and generate logical structures through rule reasoning.

**Compositional Ability of Language Model** Previous work focuses on compositionality from a linguistic viewpoint. For example, the SCAN dataset (Lake & Baroni, 2018) consists of sample pairs of natural language (e.g., "jump opposite left after walk around left") and actions (e.g., "LTURN WALK LTURN WALK LTURN WALK LTURN WALK LTURN LTURN JUMP"). SCAN serves as a task to test whether a language model can capture the elements of natural language commands. This has been extended to apply to wider settings improved the compositional generalization in semantic parsing (Kim & Linzen, 2020; Keysers et al., 2020; Jiang & Bansal, 2021; Ontañón et al., 2022). Delétang et al. (2023) and Ruoss et al. (2023) explore and improve compositional generalization in formal language settings. However, these approaches are not specifically designed for assessing compositional reasoning in models. Dziri et al. (2023) investigates the compositional reasoning ability of large language models. They used large language models like GPT-3 to explore compositionality through tasks such as elementary arithmetic problems, logic games, and dynamic programming challenges. They found that a large language model often follows pattern matching in reasoning tasks rather than systemic generalization, and the error propagates from the error of each component. While they focused on analyzing the reasoning steps and finding the causes from

|        | Training     | Prediction Acc |
|--------|--------------|----------------|
| Depth2 | Pretrained   | 1              |
| Depth3 | Pretrained   | 0.705          |
| Depth4 | Pretrained   | 0.673          |
| Depth2 | Unpretrained | 1              |
| Depth3 | Unpretrained | 0.592          |
| Depth4 | Unpretrained | 0.651          |

Table 1: The language model's boolean algebra classification generalization ability for boolean algebra. As we train the RoBERTa-base (Liu et al., 2019) with depth 1 and 2, it shows that encoder-only models have difficulty in generalizing the boolean algebra with higher depth. For the unpretrained model, we used our custom character-wise vocabulary tokenizer. We utilized a randomly initialized RoBERTa model for the unpretrained model.

it, our research concentrates on more compositionality: forming a holistic logical structure. This is because we investigated semantics and syntax based on the simpler framework of boolean logic.

## 3 BACKGROUND

Here, we introduce the boolean algebra and the philosophical motivation for selecting boolean logic for investigating the compositional reasoning ability of language models.

**Boolean algebra** Boolean algebra, also known as binary logic, is an algebra where there are only two variables, True and False, and logical operators including AND($\wedge$), OR ($\vee$), NOT ($\sim$) (Boole, 1950).[1] This two-valued logic forms the basis of computer science and the primary representation for information storage and computation in modern computer architecture. These atomic operators have clearly defined behaviours represented by truth tables where more complex expressions can be composed of simpler operations over term wrapped in parentheses. Inference requires an incremental reduction procedure, substituting complex expressions, in an ordered manner, for their equivalent values. This makes boolean algebra dataset relatively clear and easy to evaluate the generated outputs. Like other mathematical algebra, basic operations follow algebra rules such as associativity, commutativity, etc. Although in computer science and propositional logic, other operators like Exclusive OR, Logical equivalence, and Implication could be applied to boolean algebra, we only consider basic AND, OR, and NOT operators in this paper.

**Propositional logic and First-order logic** Boolean algebra is highly related to human reasoning. In the stream of history of logic, one of the notable properties of boolean logic is that the unit of the logic system is propositions. Each sentence is represented as a variable, and we can reach the conclusion by calculating based on variables. However, in contemporary first-order logic, the sentences are analyzed into two quantifiers ($\forall, \exists$), variables and predicates, which makes more varied and accurate semantics and deduction possible (Frege, 1884/1950). Logical proofs are conducted according to the formal rules between quantifiers, variables and predicates. Hence, for first-order logic, it is crucial to convert natural language sentences to logical forms by parsing and exploiting these with various logical rules. We reduce the scope in our paper and elect to focus on boolean algebras and choose not to explore whether language models analyze natural language sentences or conduct reasoning with first-order logic (Wu et al., 2022; Han et al., 2022). Saparov & He (2023) made a dataset containing a first-order logic form of natural language sentences and analyzed the reasoning ability of language models at the predicate and quantifier levels. In contrast to first-order logic, with a boolean logic system, we need not parse the sentence into the implicit structure as the unit of logic is a sentence represented as variables. Also, the compositional structures are explicitly revealed with parenthesis corresponding to logical steps, which makes boolean logic adequate for investigating compositional reasoning. The only thing that language models need to do is sequentially compute the atomic operators of the given expression based on the evident compositional structure. Hence, the aim of this paper is more elementary level than in Saparov & He (2023); to consider the

---

[1]In this paper we use the numerical notation: truth values of 0, 1 and operators $+$ for OR and $*$ for AND.

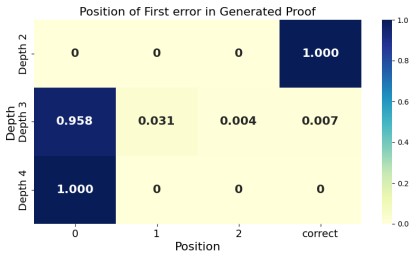

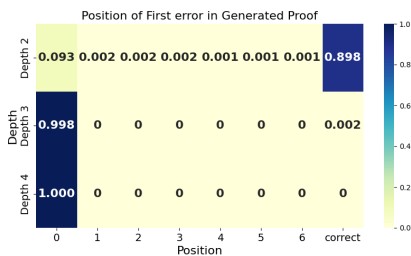

(a) Position of first error in pretrained model.

(b) Position of first error in unpretrained model.

Figure 3: Position of the first error in the generated proof on the validation set. The model was trained on depths 1 to 2, and the tokenizer for the unpretrained model is based on the pretrained tokenizer. We can see that the first error of proof is placed at earlier stage of proof in higher depth evaluation sets.

variables of boolean algebra as a form of sentences and to investigate how well language models utilize these sentences to construct a logical structure.

## 4 METHOD

In order to explore the compositional ability of models, we trained the language model with a specific boolean algebra dataset and evaluated it on more complex boolean algebra. Here, the criteria of complexity is the depth of input boolean expression.

### 4.1 DATASET FOR COMPOSITIONAL REASONING WITH BOOLEAN ALGEBRA

We can define boolean algebra as a tree structure. Let $T = (V, E, n, op)$. Nodes $v \in V$ are variables of boolean algebra, 0 or 1. Each node $v \in V$ can be assigned to function $n$ indicating unary negation function($\sim$) or $op$ which reflects binary logical operator "AND (*)" or "OR (+)". $E$ represents an argument participating in a binary function $op$.

We generated random boolean expressions for each depth without duplicates. First, we generated a random tree for each defined depth as the tree of (a) in Figure 2, then converted it to a string representation. We generated a reductive proof of the boolean expression following the *leaf nodes first rule*. First, we traversed the leaf nodes requiring calculation from left to right and computed them one by one like (a), (b), (c) in Figure 2. During the tree traversal, if we encounter basic binary calculation without negation such as (0+1) or (1*0) we resolve the nodes to one truth value like 1 or 0 ((c), (d) and (f) in Figure 2), and if we detect a negation operator, express the result in that node ((a), (b) and (e) in Figure 2). We traverse left to right then come back to the leftmost leaf and repeat again until we obtain the final truth value ((c), (d) and (f) in Figure 2) (For detailed information, please refer to Appendix B). Each tree sample of the dataset includes a boolean expression given as input to the language model, the truth value of the expression and the proof step for deriving the truth value from the original boolean expression. The input is a boolean algebra formula, which language models should solve, and the outputs should be its truth in the case of classification tasks and its proof for deriving the final answer in the generation task.

The advantage of representing boolean algebra as a tree lies in dividing the boolean algebra set according to its complexity, represented by depth. A depth 1 tree comprises only a single binary operator, which can be combined to construct more complex trees. This phenomenon demonstrates the compositionality of boolean algebra.

| | Training | Model | Prediction Acc | Exact Match | Number of matching step |
|---|---|---|---|---|---|
| Depth2 | Pretrained | T5 | 1 (±0) | 1 (±0) | 1 (±0) |
| Depth3 | Pretrained | T5 | 0.745(±0.0118) | 0.008 (±0.007) | 0.285 (±0.0056) |
| Depth4 | Pretrained | T5 | 0.686 (±0.0089) | 0 (±0) | 0.164 (±0.004) |
| Depth2 | Pretrained | GPT2 | 0.996 (±0.0044) | 0.976 (±0.0124) | 0.99 (±0.00071) |
| Depth3 | Pretrained | GPT2 | 0.648 (±0.027) | 0 (±0) | 0.216 (±0.0202) |
| Depth4 | Pretrained | GPT2 | 0.623 (±0.0134) | 0 (±0) | 0.15 (±0.0044) |
| Depth2 | Unpretrained | Seq-to-Seq | 0.949 (±0.0056) | 0.894 (±0.0041) | 0.926 (±0.0044) |
| Depth3 | Unpretrained | Seq-to-Seq | 0.539 (±0.0286) | 0.003 (±0.0026) | 0.138 (±0.0061) |
| Depth4 | Unpretrained | Seq-to-Seq | 0.518 (±0.0286) | 0 (±0) | 0.0866 (±0.0024) |
| Depth2 | Unpretrained | AutoRegressive | 0.999 (±0.0026) | 0.941 (±0.0045) | 0.982 (±0.0025) |
| Depth3 | Unpretrained | AutoRegressive | 0.608 (±0.0176) | 0.002 (±0.0025) | 0.178 (±0.0042) |
| Depth4 | Unpretrained | AutoRegressive | 0.606 (±0.0072) | 0 (±0) | 0.127 (±0.007) |

Table 2: The quantitative results of language model trained on depth 1 and 2 boolean algebra tree. Based on the quantitative metrics, it shows that language models could not learn the algorithm for solving complex boolean formulas. We use a pretrained tokenizer for Unpretrained models. We evaluated on 3 different dataset and put its mean and standard deviation.

## 4.2 EVALUATING COMPOSITIONAL GENERALIZATION IN TRANSFORMERS

**Tasks and Model Architecture**  We evaluated compositional generalization through two tasks: one is a classification task to predict the final truth value of a boolean expression, the second is a generation task that involves producing the final truth value along with its proof (Figure 2). For the classification tasks, we use the RoBERTa-base (Liu et al., 2019) configuration. For the generation task, we compare using a sequence-to-sequence model with an autoregressive model. For pre-trained models, we use T5-base (Raffel et al., 2020) and GPT-2 (Radford et al., 2019). We utilized T5-base and GPT-2 models with random weights for the unpretrained sequence-to-sequence and autoregressive models, respectively. Hyperparameter choices are listed in Appendix A. We also trained the model with several model configurations and sizes (Appendix F.5) and different tokenization configurations (Appendix F.2).

**Training**  Similar to previous generalization tasks (Clark et al., 2020; Delétang et al., 2023), all tasks were trained on specific depth levels of boolean algebra and evaluated on boolean algebra validation sets with depths exceeding those used during the training. Depth 1 was always included in the training dataset as it contains atomic formulas and makes up the compositional components of all boolean expressions. We expected the language model to learn binary and negation operators and compositional stepwise proof procedures during the training. The key insight here is that as all boolean algebra expressions are constructed using simpler boolean expressions and combination rules with parenthesis as well as logical operators and solutions to all boolean algebra expressions are derived from the application of boolean algebras with low depth levels, if the language model learns the boolean algebra dataset through compositional generalization, complex expressions can be easily generalized. In this paper, we trained the model on depth 1 and 2 or depth 1, 2 and 3 boolean expressions (Appendix 8) and validated on higher depth.

**Evaluation metrics**  For estimating the compositional generalization ability of language models, (a) We use accuracy for measuring the final truth value of the given boolean expressions; it refers to *The Number of Correct Predictions / Total Number of Validated Samples*. For the generation tasks, (b) the exact matching score (EM) means *The number of generated proof exactly corresponding with its golden proof / Total Number of Validated Samples*. However, it is a rigid metric as it requires the generated boolean solution should exactly match the golden solution. Hence, we introduce a softer metric, (c) the Number of Matching Steps, which is the average of step matching score. The definition of step matching score is *The number of steps overlapped between the generated proof and the golden proof / The length of golden proof* (See Appendix C).

|  | Correct | Lower depth tree | same depth tree | Higher depth tree | Ill-formed tree |
|---|---|---|---|---|---|
| Depth 2 | 1000 | 0 | 0 | 0 | 0 |
| Depth 3 | 7 | 562 | 92 | 0 | 339 |
| Depth 4 | 0 | 550 | 0 | 0 | 450 |

Table 3: The table displays the categorization of the first error of generated proof from the preatrained model. Since we used the model trained up to depth 2, the outcomes at depth 2 demonstrate the model's proper generalization from seen data. However, with increasing depth, we observe instances where the model generates lower-depth or ill-formed trees, ultimately resulting in incorrect proofs.

|  | Training | Model | Prediction Acc | Exact Match | Number of matching step |
|---|---|---|---|---|---|
| Depth2 | Pretrained | T5-base | 1 | 1 | 1 |
| Depth3 | Pretrained | T5-base | 0.769 | 0.016 | 0.3442 |
| Depth4 | Pretrained | T5-base | 0.741 | 0 | 0.2039 |
| Depth2 | UnPretrained | Seq-to-Seq | 0.977 | 0.939 | 0.9539 |
| Depth3 | UnPretrained | Seq-to-Seq | 0.598 | 0.002 | 0.1654 |
| Depth4 | UnPretrained | Seq-to-Seq | 0.548 | 0 | 0.1007 |

Table 4: The results of generalization with prefix representation for boolean algebra. The experiment setting is the same as boolean expression represented with parenthesis. We trained the model with depth 1 and 2 consisting of polish expression and evaluated on depth 2,3,4 polish expression dataset. We use T5-base with random parameters for unpretrained models.

## 5 RESULTS

### 5.1 COMPOSITIONAL GENERALIZATION

**Compositional reasoning generalization**   Results of the classification task are reported in Table 1. As the depth increases the model encounters more complex boolean expressions. The models did not accurately predict the truth value of boolean formulas at higher depths. The role of language model pre-training for RoBERTa was not influential: a model without pre-training demonstrated similar performance characteristics. Results of the generation task are reported in Table 2 and Table 8 in the Appendix  F.1, show similar issues in solving deeper boolean expressions. Models encounter difficulties in compositional generalization for simple boolean algebras. Both the EM score and number of matching steps decrease when the model is tested with expressions with higher depth. The models, which generate proofs almost perfectly at depth 2, exhibit poor proof generation ability at depths 3 and 4 with an EM score of near 0. These results show that the language model does not learn and apply the principle of compositionality, which could be obtained during the training on lower depths. These results align with the previous research showing that language models encounter challenges when attempting to solve more difficult datasets compared to training dataset Clark et al. (2020); Saha et al. (2021). Furthermore, it fundamentally supports and lays the foundations for the shallow reasoning ability of language models (Helwe et al., 2021) because it implies that language models cannot construct a logical structure consisting of binary values and clear logic rules before natural language sentences are introduced.

### 5.2 PROOF ANALYSIS

**Structure Recognition**   The results of the boolean logic generalization imply that the reason for the failure of systematic generalization is not only based on the various semantics and implicit deductions in natural language sentences but also on a failure to understand and construct the structure itself (Table 2). In other words, the sentence or arithmetic calculations such as "All men are mortal. Socrates is a man. Therefore, Socrates is mortal. ($\forall(Man(x) \rightarrow Mortal(x)),\ Man(Sorcates),\ Mortal(Sorcates))$)" or "$(0+1) \rightarrow 1$" might not the only obstacles in the reasoning of language models. Figure  3 and Table  3 outline the reason for this failure. In Figure 3, both the pretrained and unpretrained models yield high scores for shallow depths. However, from the depth 3 evaluation set, both models showed a high rate of generating wrong boolean expressions at index 0, with percentages of $0.958$ and $0.998$ respectively. For deeper trees it was

evident that both models started generating errors from the beginning of the proof. From Figure 3, we can observe that the error in index 0 is due to generating lower depth and ill-formed tree compared to golden step. When language models encounter a more complex tree, the error appears in the early stage, and most errors stem not from calculations, but from incorrect structure recognition and generation. The low scores of exact match and step matching arise from the generation of these incorrect structures. In other words, the language model cannot represent and generate unseen structures. Considering that understanding compositional rules makes it easy to comprehend complex structures, it suggests that the language models do not learn according to boolean compositional rules and they recognize complex boolean expressions as out-of-distribution data. The implication of this experimental result in a natural language setting is that, even if the training data presents new problems which do not include new concepts, new definitions and formulas, and other creative proofs that were never seen before, but only include more proof steps, the language model might not solve them in a step-by-step way, with relying solely on the trained knowledge, and it makes step-by-step rationale unreliable.

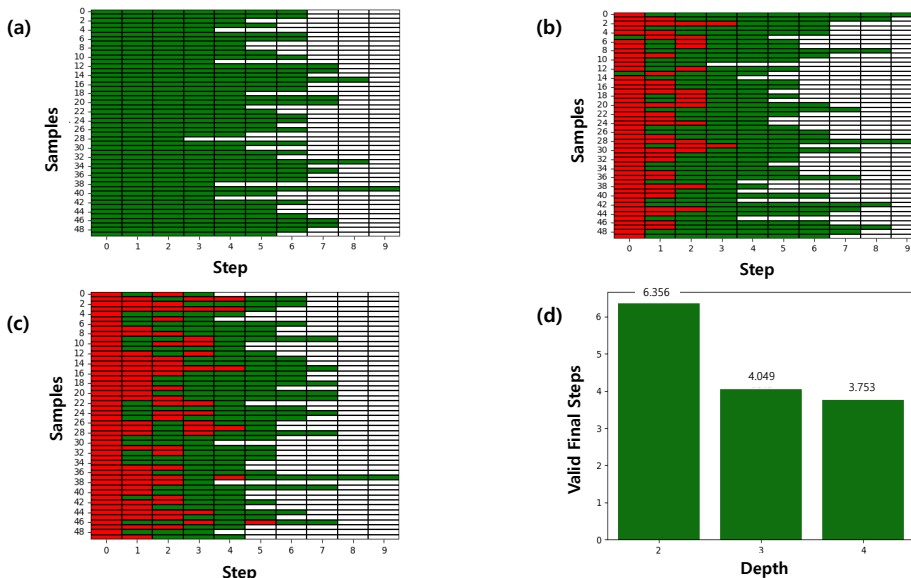

Figure 4: The analysis for generated proof from the model trained on depth 1 to 2 boolean expression. (a) is the result for the validation dataset containing depth 2 boolean expression, (b) and (c) are the results of depth 3 and depth 4 each. As the y-axis is an index of the generated proof and the Green item in (a), (b), (c) represent the valid calculation between the two step, we can observe that in complex boolean expression, the model generate valid deductions. (d) is the average number of the last consecutive, valid calculations of generated proof.

**How do language models recover the answer from the wrong proof?** Assessing the generated proof is still challenging for both machines and humans. Although there are some fine-grained methods to evaluate the steps (Golovneva et al., 2023; Prasad et al., 2023), the current trend for measuring the reasoning performance of language models is identifying the model's final prediction in generated outputs (Wang et al., 2023; Yu et al., 2023). In this direction, one strange finding is that the models return to the correct answer from the wrong proof (Saparov & He, 2023). We couldn't determine whether the language model generates the final answer entirely independent of the proof or if it corrects itself at some point and produces the right proof step. Through the boolean algebra parser, we can verify compositionality or calculate correctness between two steps and found that the final answer is not independent of the proof and the last few steps of proof succeed in the correct calculation compositionally, the generated outputs have an average $4.049$ and $3.753$ final consecutive valid steps in depth 3 and 4 evaluation dataset, which leads to generating the final answer (Figure 4). However, Figure 3 and Table 3 show that the earlier steps of proof contain errors, resulting in language models being right for the wrong reason.

**Polish notation for boolean expression**    Like in natural language, where sentences with the same meaning can be expressed differently, Boolean algebra can also be represented in various notations. To investigate if the notation affects compositional ability, we converted all boolean expressions from the conventional infix notation to Polish notation and conducted the same experiment. Table 4 shows that with Polish notation, the performance increases in prediction and proof scores except for the Exact Match of un-pretrained model at depth 3. As Polish notation represents the formula without parenthesis and determines the order of operation only through the position of operators and variables, the language model does not hallucinate due to the parenthesis. However, despite the positive aspects in terms of computational efficiency or character efficiency, the language model still fails to address the generalization issue with respect to depths.

**Stepwise Generation**    In Yang et al. (2022) and Zhou et al. (2023), for enhancing the validity of each step, prompts for generating fine-grained rationales or verifiers for individual steps have been introduced to improve the quality of overall rationales. In this aspect, the key is to guide the language model to generate each step accurately and recursively complete the entire proof. We conducted stepwise experiments (see Appendix D) where results failed to generalize complex structures like all-at-once generation. We believe this demonstrates the importance of a holistic understanding of proof structure and planning in forming a rationale that progresses from the input to the correct output.

**Length of tree**    In spite of the results in Table 1 and Table 2, decreased performance in higher depth could be affected by the length of boolean expressions, not depth. To handle this issue, we evaluate the trained model on a specific dataset where the length of boolean expression inputs is similar to strings in a training dataset. Specifically, we removed all strings in depth 2 and depth 3 validation sets longer than the longest string in the training dataset, and for depth 4, we allowed up to the length of the longest string in the training dataset plus 5 due to dataset size constraints. The result in Table 7 in Appendix E shows a similar trend with previous experiments. We can also observe that the performance drops more steeply when increasing the depth rather than increasing the length in Appendix E. These results demonstrate that the inability to solve compositional tasks is not solely due to input length.

## 6    DISCUSSION AND CONCLUSION

It is critical that language models accurately represent compositional structures rather than exploit memorized patterns in language. We empirically demonstrated that language models lack the ability for compositional generalization through boolean expressions, resulting in a failure to construct logical structures. We observed that by leveraging the explicit compositional feature of boolean expressions, the transformer architecture not only fails to perform compositional reasoning on boolean expression problems due to its inability to understand the input systematically but also recognizes complex boolean expressions as an entirely different problem. The implications of our findings for natural language reasoning tasks suggest that recent advances in reasoning are not results of language models systematically understanding compositional structures in text. Instead, it implies that when language models are faced with actual out-of-distribution problems as well as pseudo-out-of-distribution problems like complex boolean expressions, they lose their reliability.

## 7    LIMITATIONS & FURTHER RESEARCH

We selected a formal language, rather than a natural language, to assess the logical structural compositionality ability of the language model. However, there's an argument that natural language and rigorous logic inhabit entirely different languages (Wittgenstein, 1953; Grice, 1975). Therefore, we must also attempt similar experiments in a natural language setting like whether step-by-step reasoning can be used to solve compositional reasoning challenges in large-scale models. Previous studies have focused on improving the generalization ability for formal languages based on Chomsky hierarchy through positional encoding or recursive networks. However, systematic generalization in reasoning is a different field. In order to enhance systematic reasoning, we might need a different architecture or planning algorithm for planning and holistic awareness of tasks, not relying on conventional attention and positional encoding.

## 8 REPRODUCIBILITY

We used RoBERTa, T5 and GPT2 model from huggingface library, and PyTorch lightning for training (for detailed information, please refer to Appendix A). For API experiments in Appendix F.4, we used OpenAI API on September 27th, 28th, 2023. We attach part of the code and our datasets and will publish our code and datasets in the future.

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

APPENDIX

## A EXPERIMENT DETAILS

We used T5-base and GPT2 for generation tasks and RoBERTa-base for classification tasks. For proof analysis, we used generated samples from pretrained T5-base model trained at depth 1 and 2 boolean expressions. We used AdamW for optimizing (Loshchilov & Hutter, 2019) and set epoch 100 for both tasks. We use a learning rate of 0.0001 and batch size 64 for RoBERTa and a learning rate 0.00005, batch size 16 for T5-base and GPT2. All models were trained on either A6000, A4000 or A100 NVIDIA GPUs.

We can generate random boolean expressions, but we used a train dataset containing 64 depth 1 trees and 16,420 depth 2 trees, summing up to 16,484 for depth 1,2 training. The second train dataset contains 64 depth 1 boolean expressions, 10,000 depth 2, and 20,000 samples for depth 3, totalling 30,064 samples. The validation set for each depth consists of 1,000 samples.

## B ALGORITHM FOR PROOF GENERATION

---
**Algorithm 1** Tree parsing algorithm
---
**Require:** Input is binary tree $T = (V, E, n, op)$ defined in section 4.1
**Ensure:** Root node where the value is 0 or 1
  $proof\_list \leftarrow empty\ list$
  $leaves\_list \leftarrow leaves\ of\ input\ tree$
  **while** $len(leaves\_list)! = 1$ **do**
    $length \leftarrow len(leaves\_list)$
    $i \leftarrow 0$
    **while** $i < length$ **do**
      **if** $leaves\_list\,[i]\,.n\ exist$ **then**
        $Calculated\_result \leftarrow n(leaves\_list\,[i])$
        $Replace\_unary(leaves\_list, leaves\_list\,[i]\,, Calculated\_result)$     ▷ Replace v in
leave_list to Calculated result
        $i \leftarrow i + 1$
      **else if** $leaves\_list\,[i]\,.op\ exist$ **then**
        $Calculated\_result \leftarrow op(leaves\_list\,[i]\,, leaves\_list\,[i+1])$
        $Replace\_binary(leaves\_list, leaves\_list\,[i]\,, leaves\_list\,[i+1]\,, Calculated\_result)$
  ▷ Replace v and next of v in leaves_list to Calculated result
        $i \leftarrow i + 2$
      **end if**
      $proof\_list.append(leaves\_list)$
    **end while**
    **if** $len(leaves\_list) == 1\ and\ v.n\ not\ exist$ **then**
      $break$
    **end if**
  **end while**
---

## C EVALUATION METRICS

Our three evaluation metrics are based on matching between steps in the gold proof and the generated proof. The application examples can be found in Table 5 and Table 6

**Prediction Accuracy**   In a classification task, we train the model to classify whether the truth value of a given input is 0 or 1. In generation tasks, prediction Accuracy is determined by comparing the last element of the golden proof with the last element of the generated proof. If they match, 1 is assigned; otherwise, it is 0.

**Exact Match**  Exact match involves a one-to-one comparison of all elements in the golden proof with those in the generated proof. If the two proofs match perfectly, a 1 is given; otherwise, it is 0.

**Number of matching step**  It refers to the number of elements shared between the gold proof and the generated proof. In other words, it's the size of the intersection between Set(Gold proof) and Set(Generated proof).

| | |
|---|---|
| **Gold proof** | ["∼(∼1*∼0)", "∼(∼1*1)", "∼(0*1)", "∼0", "1"] |
| **Generated Proof** | ["∼(∼1*∼0)", "∼(∼1*1)", "∼(0*1)", "∼0", "1"] |
| **Prediction Accuracy** | 1 |
| **Exact Match** | 1 |
| **Number of Matching Step** | 1 |

Table 5: The example of evaluation metric. As the gold proof and generated proof are exactly the same, Prediction Accuracy, Exact Match, and Number of Matching Stpe are all 1.

| | |
|---|---|
| **Gold proof** | ["∼(∼1*∼0)", "∼(∼1*1)", "∼(0*1)", "∼0", "1"] |
| **Generated Proof** | ["∼(∼1*∼0)", "∼(∼1*1)", "(0*1)", "0", "1"] |
| **Prediction Accuracy** | 1 |
| **Exact Match** | 0 |
| **Number of Matching Step** | 0.6 |

Table 6: The example of evaluation metric. As the gold proof and generated proof have same last element, Prediction Accuracy is 1, however, as third and fourth elements are different, Exact Match is 0 and the Matching Step score is 0.6.

## D    STEPWISE GENERATION

Stepwise generation involves training the language model to generate each step one at a time, using the given boolean expression and partial proof as context. During inference, this process is recursively repeated from the first step to generate the entire proof. The result appears in Figure 5.

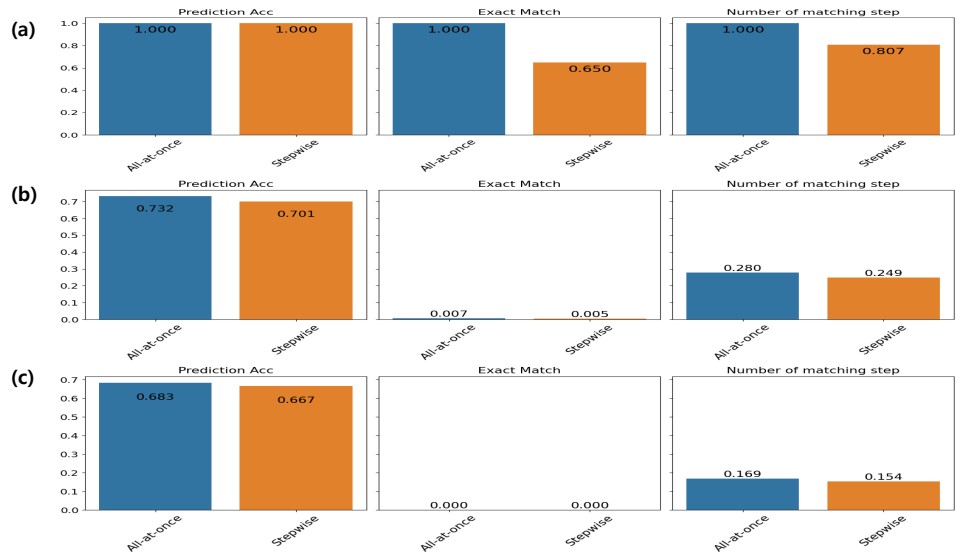

Figure 5: The comparison between generating proof at once and stepwise generation. (a), (b), and (c) represent the results of the depth 2, 3, and 4 evaluation sets, respectively.

# E    LENGTH ABLATION

We compared two models. One was trained on a dataset containing boolean question expressions with a specific maximum length of 28, using the T5-base model. We then evaluated it on a validation set containing boolean expression length 29, 30, 31, 32 and 33. The other model was trained on a dataset containing regular boolean expressions with depths of 1 to 2, also using the T5-base model. It was evaluated at depths 2, 3, and 4. The result shows in Figure  6

|        | Training     | Prediction Acc | Exact Match | Number of matching step |
|--------|--------------|----------------|-------------|-------------------------|
| Depth2 | Pretrained   | 1              | 1           | 1                       |
| Depth3 | Pretrained   | 0.7928         | 0.01934     | 0.4131                  |
| Depth4 | Pretrained   | 0.7607         | 0           | 0.2614                  |
| Depth2 | UnPretrained | 0.9788         | 0.9495      | 0.9701                  |
| Depth3 | UnPretrained | 0.5801         | 0           | 0.1917                  |
| Depth4 | UnPretrained | 0.5276         | 0           | 0.1174                  |

Table 7: The table shows the results of pruned evaluation dataset. As the pruned dataset contains a tree with a similar length to the train dataset, it shows that the language model's poor composition ability is highly attributed to the depth of boolean algebra. We used T5-base model for pretrained model and random weighted T5-base model for unpretraiend

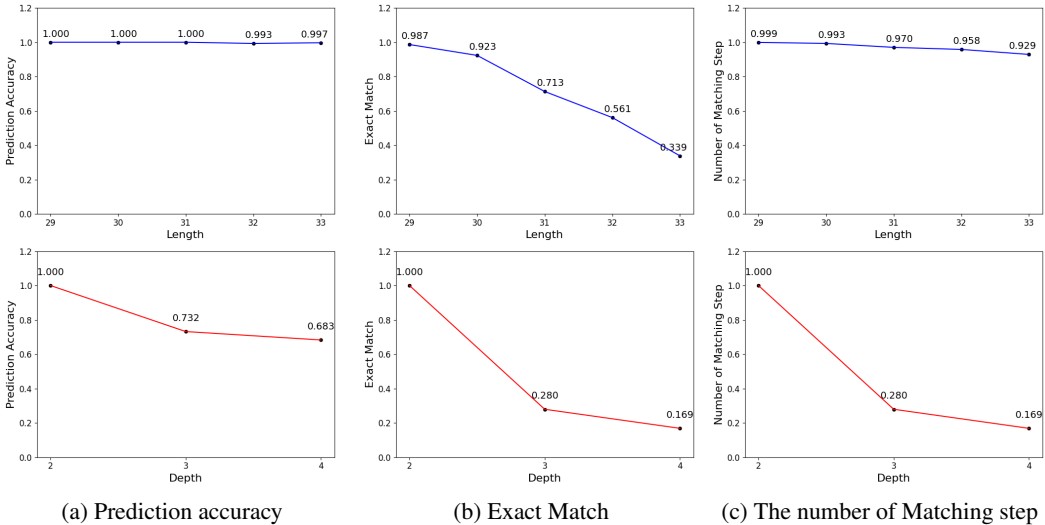

(a) Prediction accuracy     (b) Exact Match     (c) The number of Matching step

Figure 6: The above blue graph illustrates the performance decrease with increasing length, while the bottom graph depicts the performance change with increasing depth. We can see that the performance gap widens more significantly with an increase in depth compared to an increase in length. The model was trained on depths 1 to 2 and it is a pretrained model.

# F    ADDITIONAL EXPERIEMNT RESULTS

## F.1    GENERALIZATION EXPERIMENTS

We conducted generalization experiments training with depth 1,2 and 3 boolean trees. In comparison to the training on depth 1 and 2, the model exhibited better performance on the validation set with one additional depth. However, for more complex datasets like depth 5 and 6, it showed a lack of generalization ability. We can see the results in Table  8, which has the same maximum output length is 256 and Table  9 which has 1024 maximum output length. The trend of a significant decrease with increasing depth remains consistent as the language model tends to produce errors from the first when dealing with complex data.

|  | Training | Model | Prediction Acc | Exact Match | Number of matching step |
|---|---|---|---|---|---|
| Depth3 | Pretrained | T5 | 0.978 (±0.0047) | 0.978 (±0.0047) | 0.993 (±0.0019) |
| Depth4 | Pretrained | T5 | 0.786 (±0.0055) | 0.289 (±0.0155) | 0.608 (±0.0032) |
| Depth5 | Pretrained | T5 | 0.578 (±0.0297) | 0.015 (±0.0055) | 0.232 (±0.0116) |
| Depth6 | Pretrained | T5 | 0.459 (±0.0116) | 0 (±0) | 0.106 (±0.0013) |
| Depth3 | Pretrained | GPT2 | 0.998 (±0.0012) | 0.984 (±0.002) | 0.993 (±0.0003) |
| Depth4 | Pretrained | GPT2 | 0.85 (±0.0108) | 0.174 (±0.0102) | 0.486 (±0.0116) |
| Depth5 | Pretrained | GPT2 | 0.721 (±0.0207) | 0.002 (±0.002) | 0.192 (±0.0083) |
| Depth6 | Pretrained | GPT2 | 0.679 (±0.0182) | 0 (±0) | 0.12 (±0.0081) |
| Depth3 | UnPretrained | Seq-to-Seq | 0.957 (±0.0078) | 0.902 (±0.0102) | 0.9363 (±0.0047) |
| Depth4 | UnPretrained | Seq-to-Seq | 0.621 (±0.008) | 0.007 (±0.002) | 0.196 (±0.0009) |
| Depth5 | UnPretrained | Seq-to-Seq | 0.444 (±0.0211) | 0 (±0) | 0.092 (±0.0048) |
| Depth6 | UnPretrained | Seq-to-Seq | 0.286 (±0.1118) | 0 (±0) | 0.042 (±0.0031) |
| Depth3 | UnPretrained | AutoRegressive | 0.996 (±0.0012) | 0.77 (±0.0095) | 0.968 (±0.0023) |
| Depth4 | UnPretrained | AutoRegressive | 0.763 (±0.0068) | 0.014 (±0.004) | 0.263 (±0.0061) |
| Depth5 | UnPretrained | AutoRegressive | 0.702 (±0.0081) | 0 (±0) | 0.1524 (±0.0052) |
| Depth6 | UnPretrained | AutoRegressive | 0.688 (±0.0082) | 0 (±0) | 0.114 (±0.0024) |

Table 8: The generalization result of language model trained on depth 1, 2 and 3 boolean algebra tree. We used pretrained tokenizer for unpretrained model as well. We put the mean and standard deviation of 3 different evaluation sets as well. We used a randomized weight T5-base and GPT2 model for unpretrained models.

|  | Training | Model | Prediction Acc | Exact Match | Number of matching step |
|---|---|---|---|---|---|
| Depth3 | Pretrained | T5 | 0.979 (±0.0044) | 0.979 (±0.0044) | 0.993 (±0.0017) |
| Depth4 | Pretrained | T5 | 0.773 (±0.0067) | 0.271 (±0.0245) | 0.589 (±0.0072) |
| Depth5 | Pretrained | T5 | 0.575 (±0.0203) | 0.011 (±0.0047) | 0.222 (±0.0093) |
| Depth6 | Pretrained | T5 | 0.459 (±0.0087) | 0 (±0) | 0.11 (±0.0007) |
| Depth3 | UnPretrained | Seq-to-Seq | 0.929 (±0.0025) | 0.863 (±0.0176) | 0.914 (±0.0115) |
| Depth4 | UnPretrained | Seq-to-Seq | 0.618 (±0.0053) | 0.008 (±0.0012) | 0.197 (±0.0024) |
| Depth5 | UnPretrained | Seq-to-Seq | 0.421 (±0.0143) | 0 (±0) | 0.089 (±0.0011) |
| Depth6 | UnPretrained | Seq-to-Seq | 0.272 (±0.0106) | 0 (±0) | 0.041 (±0.0021) |

Table 9: The generalization result of language model trained on depth 1, 2 and 3 boolean algebra tree with maximum output length 1024. We used pretrained tokenizer for unpretrained model as well. We put the mean and standard deviation of 3 different evaluation sets as well. We used a randomized weight T5-base model for unpretrained models.

### F.2 CHARACTERWISE TOKENIZER

Instead of using pretrained tokenizer for training unpretrained model, we can use a characterwise tokenizer where each token corresponds to each character. For example, "(" is assigned to 7 and "0" is assigned to 13. We can also observe the performance decrease as we evaluate the model in more complex depth. We can also see slight decrease in performance compared to the model trained with pretrained tokenizer. We attribute this to using a characterwise tokenizer, which we believe resulted in a slight performance drop compared to when a pretrained tokenizer was used. This is because the number of tokens that need to be processed increases when we use a characterwise tokenizer. For instance, in a pretrained tokenizer, one token id corresponds to '+1', whereas in an characterwise setting, '+' and '1' have different token ids.

|        | Training    | Tokenizer     | Model      | Prediction Acc | Exact Match | Number of matching step |
|--------|-------------|---------------|------------|----------------|-------------|-------------------------|
| Depth2 | UnPretrained | Characterwise | Seq-to-Seq | 0.979          | 0.95        | 0.9704                  |
| Depth3 | UnPretrained | Characterwise | Seq-to-Seq | 0.531          | 0           | 0.1292                  |
| Depth4 | UnPretrained | Characterwise | Seq-to-Seq | 0.512          | 0           | 0.0834                  |

Table 10: The result of unpretrained T5-base with characterwise tokenizer which is trained with depth 1 and 2 boolean expressions. We used a T5-base model with random parameters for unpretrained models.

|        | Training    | Tokenizer     | Model      | Prediction Acc | Exact Match | Number of matching step |
|--------|-------------|---------------|------------|----------------|-------------|-------------------------|
| Depth3 | UnPretrained | Characterwise | Seq-to-Seq | 0.926          | 0.815       | 0.8652                  |
| Depth4 | UnPretrained | Characterwise | Seq-to-Seq | 0.577          | 0.012       | 0.1866                  |
| Depth5 | UnPretrained | Characterwise | Seq-to-Seq | 0.372          | 0           | 0.07925                 |
| Depth6 | UnPretrained | Characterwise | Seq-to-Seq | 0.186          | 0           | 0.0394                  |

Table 11: The result of unpretrained T5-base with characterwise tokenizer which is trained with depth 1, 2 and 3 boolean expressions. We used a T5-base model with random parameters for unpretrained models.

## F.3 TREE-WEIGHT EXPERIMENT

|  | Subtree | Prediction Acc | Exact Match | Number of matching step |
|---|---|---|---|---|
| Depth3 | Left | 1 | 1 | 1 |
| Depth4 | Left | 0.8775 | 0.2779 | 0.6013 |
| Depth5 | Left | 0.6906 | 0.01121 | 0.2237 |
| Depth6 | Left | 0.6609 | 0 | 0.1344 |
| Depth3 | Right | 1 | 1 | 1 |
| Depth4 | Right | 0.9495 | 0.4183 | 0.7163 |
| Depth5 | Right | 0.8366 | 0.02151 | 0.2783 |
| Depth6 | Right | 0.7637 | 0 | 0.1502 |
| Depth3 | Balanced | 0.9694 | 0.7249 | 0.9531 |
| Depth4 | Balanced | 0.4646 | 0.007 | 0.1989 |
| Depth5 | Balanced | 0.3371 | 0 | 0.0543 |
| Depth6 | Balanced | 0.2169 | 0 | 0.02073 |

Table 12: This presents the generalization results based on left or right weighted trees. We trained the T5-base pretrained model on the same depth 1-3 datasets (same as Table 8 and divided the validation set into sets with left-weighted subtrees, right-weighted subtrees, and balanced.

As a boolean expression could be represented as a binary tree, we can divide the boolean algebra into left-weighted, where the left subtree has more depth than the right subtree, and right-weighted where the right subtree is deeper than the left subtree. If the depth of the left subtree is greater, the language model needs to perform more calculations towards the front in the early stages of the proof. Conversely, if the depth of the right subtree is greater, it should focus on computations in the later part of the expression in the early stages of proving. The results show that the right-weighted subtree shows better performance than other subtree (See Table 12).

## F.4 EXPERIMENT ON LARGE-SCALE LANGUAGE MODEL

We evaluate proof generation ability of GPT-3.5-turbo and text-davinci-003. We use 8-shot in-context learning, which means the language model experiences 8 examples of our boolean expression and proof procedures (See Table 13). Even though it is simple axiom-based reasoning compared to problems recent large language models tackle, but learning our proof algorithm through in-context learning has proven to be challenging. (Table 14) shows that large-language models cannot generalize the examples. Since large language models already tend to generate proofs in a specific manner, we speculate that our algorithm is difficult to learn through a few examples.

| | |
|---|---|
| **Prompt** | **Question**: "∼((∼(∼1*1)*1)+∼(∼1+(0*∼0)))",
**Answer**: ["∼((∼(0*1)*1)+∼(∼1+(0*∼0)))", "∼((∼(0*1)*1)+∼(0+(0*∼0)))",
"∼((∼(0*1)*1)+∼(0+(0*1)))",
"∼((∼0*1)+∼(0+(0*1)))", "∼((∼0*1)+∼(0+0))",
"∼((1*1)+∼(0+0)", "∼((1*1)+∼0)", "∼(1+∼0)", "∼(1+1)", "∼1", "0"]

**Question**: "(∼(∼1+1)+∼(∼0*∼1))",
**Answer**: ["(∼(0+1)+∼(∼0*∼1))", "(∼(0+1)+∼(1*∼1))", "(∼(0+1)+∼(1*0))",
"(∼1+∼(1*0))", "(∼1+∼0)", "(0+∼0)", "(0+1)", "1"]

**Question**: "∼(∼0+(∼(0*∼1)+0))",
**Answer**: ["∼(1+(∼(0*∼1)+0))", "∼(1+(∼(0*0)+0))",
"∼(1+(∼0+0))", "∼(1+(1+0))", "∼(1+1)", "∼1", "0"]

**Question**: "(1*((0+∼1)+(∼0*1)))",
**Answer**: ["(1*((0+0)+(∼0*1)))", "(1*((0+0)+(1*1)))",
"(1*(0+(1*1)))", "(1*(0+1))", "(1*1)", "1"]

**Question**: "((∼(∼0*1)*∼(1*1))+0)",
**Answer**: ["((∼(1*1)*∼(1*1))+0)", "((∼(1*1)*∼1)+0)",
"((∼1*∼1)+0)", "((∼1*0)+0)", "((0*0)+0)", "(0+0)", "0"]

**Question**: "((∼1+∼0)+∼(0+0))",
**Answer**: ["((0+∼0)+∼(0+0))", "((0+1)+∼(0+0))", "((0+1)+∼0)", "(1+∼0)", "(1+1)", "1"]

**Question**: "(∼(∼(1*∼0)*(1+1))*(∼(0*∼0)*(0*1)))",
**Answer**: ["(∼(∼(1*1)*(1+1))*(∼(0*∼0)*(0*1)))", "(∼(∼(1*1)*1)*(∼(0*∼0)*(0*1)))",
"(∼(∼(1*1)*1)*(∼(0*1)*(0*1)))",
"(∼(∼(1*1)*1)*(∼(0*1)*0))", "(∼(∼1*1)*(∼(0*1)*0))", "(∼(∼1*1)*(∼0*0))",
"(∼(0*1)*(∼0*0))", "(∼(0*1)*(1*0))", "(∼0*(1*0))", "(∼0*0)", "(1*0)", "0"]

**Question**: "∼(∼(∼(∼0+1)+(1*0))+∼1)",
**Answer**: ["∼(∼(∼(1+1)+(1*0))+∼1)", "∼(∼(∼(1+1)+0)+∼1)",
"∼(∼(∼(1+1)+0)+0)", "∼(∼(∼1+0)+0)",
"∼(∼(0+0)+0)", "∼(∼0+0)", "∼(1+0)", "∼1", "0"]

**Question**:"(0*∼(∼(0+0)*∼(∼1*1)))" |

Table 13: Large-scale Language Model Evaluation

| | Model | Prediction Acc | Exact Match | Number of matching step |
|---|---|---|---|---|
| Depth3 | gpt-3.5-turbo | 0.64 | 0.02 | 0.2767 |
| Depth4 | gpt-3.5-turbo | 0.54 | 0 | 0.6013 |
| Depth5 | gpt-3.5-turbo | 0.42 | 0.01 | 0.1092 |
| Depth6 | gpt-3.5-turbo | 0.24 | 0 | 0.0559 |
| Depth3 | text-davinci-003 | 0.66 | 0 | 0.3377 |
| Depth4 | text-davinci-003 | 0.68 | 0 | 0.2115 |
| Depth5 | text-davinci-003 | 0.46 | 0 | 0.1164 |
| Depth6 | text-davinci-003 | 0.44 | 0 | 0.0876 |

Table 14: The results of generation tasks in gpt-3.5-turbo and text-davinci-003. The number of samples is 100, which are from the original validation set. We used temperature=0.5, max_tokens=512, top_p=1.0, frequency_penalty=0.0, presence_penalty=0.6.

## F.5 GRID SEARCH

To understand which configuration of the language model correlates with Boolean expression understanding, we conducted a grid search. The targeted hyperparameters included the dimension of the hidden state, the number of layers, the dimension of the feedforward network, and the number of attention heads. We used a character-wise tokenizer for RoBERTa-base and a pretrained tokenizer for T5-base. The combination of configurations is represented in Table 15 and Table 16. The results in Table 17 and Table 18 show that for all depth of validation sets, the dimension of hidden state shows a positive correlation with the performance.

| Dimension of Hidden State | [64, 128, 512, 1024] |
|---|---|
| **Number of layers** | [4, 8, 12, 16, 20, 24] |
| **Dimension of FeedForward network** | [256, 512, 1024, 2048, 4096] |
| **Number of Attention heads** | [8, 16, 32] |

Table 15: The hyperparameter configurations for RoBERTa model.

| Dimension of Hidden State | [64, 128, 768, 1024] |
|---|---|
| **Number of layers** | [4, 12, 16] |
| **Dimension of FeedForward network** | [256, 1024, 3072] |
| **Number of Attention heads** | [8, 12, 16] |

Table 16: The hyperparameter configurations for T5 model.

|  | Dimension of Hidden State | Num_layers | Num_heads | D_ff |
|---|---|---|---|---|
| Depth2 | 0.300 | 0.314 | 0.101 | 0.057 |
| Depth3 | 0.729 | -0.095 | 0.183 | 0.269 |
| Depth4 | 0.456 | 0.031 | 0.079 | 0.071 |

Table 17: We trained the unpretrained RoBERTa model for each configuration combination and evaluated on each validation set. It shows the linear correlation between prediction accuracy and each hyperparameter.

|  | Dimension of Hidden State | Num_layers | Num_heads | D_ff |
|---|---|---|---|---|
| Depth2 | 0.729 | -0.095 | 0.183 | 0.269 |
| Depth3 | 0.456 | 0.031 | 0.079 | 0.071 |

Table 18: We trained the unpretrained T5 model for each configuration combination and evaluated on each validation set. It shows the linear correlation between Number of matching step and each hyperparameter.

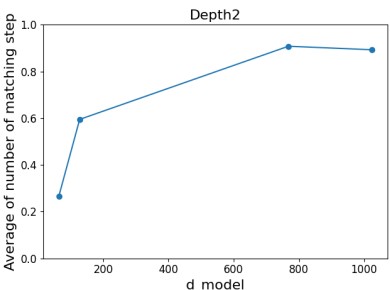 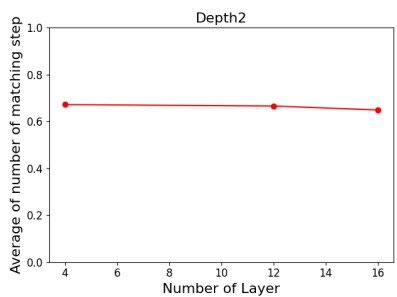

(a) Matching steps according to d_model.    (b) Matching steps according to num_layer.

Figure 7: The results show that when evaluating on depth 2 dataset. y-axis represents the average of number of matching step scores. The dimension of the hidden state has a positive effect on the model performance, but the number of layers does not.

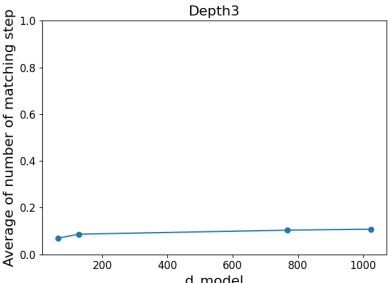 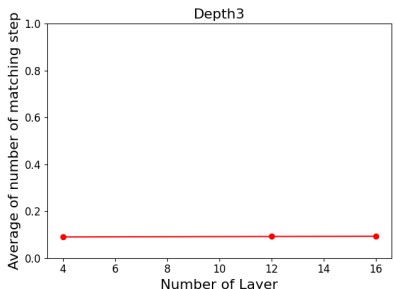

(a) Matching steps according to d_model.    (b) Matching steps according to num_layer.

Figure 8: The results show that when evaluating on depth 3 dataset. y-axis represents the average of number of matching step scores. The dimension of the hidden state has a little positive effect on the model performance, but the number of layers does not.

## G POLISH NOTATION SAMPLES

Unlike infix notation, which uses parenthesis to indicate the order of operations and places operators between operands, Polish notation explicit the order of operations by placing the operator directly in front of the operands. An example can be found in the Table 19.

| Original Question | "(1*∼(∼0*(∼1*∼1)))" |
|---|---|
| Original Proof | ["(1*∼(1*(∼1*∼1)))", "(1*∼(1*(0*∼1)))", "(1*∼(1*(0*0)))", "(1*∼(1*0))", "(1*∼0)", "(1*1)", "1"] |
| Polish Notation for Question | "*1∼*∼0*∼1∼1" |
| Polish Notation for Proof | ["*1∼*1*∼1∼1", "*1∼*1*0∼1", "*1∼*1*00", "*1∼*10", "*1∼0", "*11", "1"] |

Table 19: The generated proof in depth 3 validation set, which shows correct proof steps and final prediction.

## H ERROR SAMPLES

We present actual output samples generated by the model in Table 20, Table 21 and Table 22

| Question | "∼(∼((∼0*∼0)+∼1)*((1*∼0)+∼0))" |
|---|---|
| Generated proof | ["∼(∼((∼0*∼0)+∼1)*((1*∼0)+∼0))",
"∼(∼((1*∼0)+∼1)*((1*∼0)+∼0))",
"∼(∼((1*1)+∼1)*((1*∼0)+∼0))",
"∼(∼((1*1)+0)*((1*∼0)+∼0))",
"∼(∼((1*1)+0)*((1*1)+∼0))",
"∼(∼((1*1)+0)*((1*1)+1))",
"∼(∼(1+0)*((1*1)+1))",
"∼(∼(1+0)*(1+1))",
"∼(∼1*(1+1))",
"∼(∼1*1)",
"∼(0*1)",
"∼0",
"1"] |
| Target proof | ["∼(∼((∼0*∼0)+∼1)*((1*∼0)+∼0))",
"∼(∼((1*∼0)+∼1)*((1*∼0)+∼0))",
"∼(∼((1*1)+∼1)*((1*∼0)+∼0))",
"∼(∼((1*1)+0)*((1*∼0)+∼0))",
"∼(∼((1*1)+0)*((1*1)+∼0))",
"∼(∼((1*1)+0)*((1*1)+1))",
"∼(∼(1+0)*((1*1)+1))",
"∼(∼(1+0)*(1+1))",
"∼(∼1*(1+1))",
"∼(∼1*1)",
"∼(0*1)",
"∼0",
"1"] |

Table 20: The generated proof in depth 3 validation set, which shows correct proof steps and final prediction.

| Question | "((0+∼(1*∼(1+∼1)))*1)" |
|---|---|
| **Generated proof** | ["((0+∼(1*∼(1+∼1)))*1)",
"((0+∼(1*∼(1+0)))*1)",
"((0+∼(1*∼1))*1)",
"((0+∼1)*1)",
"((0+0)*1)",
"(0*1)",
"0"] |
| **Target proof** | ["((0+∼(1*∼(1+∼1)))*1)",
"((0+∼(1*∼(1+0)))*1)",
"((0+∼(1*∼1))*1)",
"((0+∼(1*0))*1)",
"((0+∼0)*1)",
"((0+1)*1)",
"(1*1)",
"1"] |

Table 21: The generated proof in depth 4 validation set, which shows incorrect proof step and final prediction.

| Question | "(∼(((0*∼(∼(∼0*1)+(1+1)))*∼0)+0)*∼0)" |
|---|---|
| **Generated proof** | ["(∼(((0*∼(∼(∼0*1)+(1+1)))*∼0)+0)*∼0)",
"(∼((0*∼(∼(1*1)+(1+1)))*∼0)",
"(∼((0*∼(∼(1*1)+1))*∼0)*∼0)",
"(∼((0*∼(∼(1*1)+1))*1)*∼0)",
"(∼((0*∼(∼(1*1)+1))*1)",
"(∼((0*∼1)*1)*1)",
"(∼((0*0)*1)",
"(∼(0*1)*1)",
"(∼0*1)",
"(1*1)",
"1"] |
| **Target proof** | ["(∼(((0*∼(∼(∼0*1)+(1+1)))*∼0)+0)*∼0)",
"(∼(((0*∼(∼(1*1)+(1+1)))*∼0)+0)*∼0)",
"(∼(((0*∼(∼(1*1)+1))*∼0)+0)*∼0)",
"(∼(((0*∼(∼(1*1)+1))*1)+0)*∼0)",
"(∼(((0*∼(∼(1*1)+1))*1)+0)*1)",
"(∼(((0*∼(∼1+1))*1)+0)*1)",
"(∼(((0*∼(0+1))*1)+0)*1)",
"(∼(((0*∼1)*1)+0)*1)",
"(∼(((0*0)*1)+0)*1)",
"(∼((0*1)+0)*1)",
"(∼(0+0)*1)",
"(∼0*1)",
"(1*1)",
"1"] |

Table 22: The generated proof in depth 6 validation set, which shows wrong proof step but correct final prediction.

