# OpenReview forum: "On Compositional Generalization in Language Models"
_ICLR.cc/2024/Conference — ICLR 2024 Conference Withdrawn Submission_

### Official Review · Reviewer_dEnH · 2023-10-30

**Soundness:** 2 fair
**Presentation:** 2 fair
**Contribution:** 1 poor
**Rating:** 3
**Confidence:** 4

**Summary:**

The authors create a dataset of boolean expressions, and feed it to LLMs to get the value for the boolean expression, along with the steps used to arrive at that value i.e. the proof. For example, the input expression $(((1 \land \neg 0 ) \lor \neg 1)\lor 1)$ should give an output of a set of strings resolving the innermost bracket in each step, finally giving out the final value. In this case, the expected output is $((1 \lor \neg 1)\lor 1)$, $(1\lor 1)$, 1.  The authors aim to understand LLM's compositional generalization by understanding their performance on such examples.

**Strengths:**

Clear presentation

**Weaknesses:**

- Originality: The idea is somewhat original, in testing LLMs generalization capacity through boolean expressions.

- Significance: The results have minimal to no impact on current understanding of LLMs. The boolean expressions proposed by authors are not some interesting fragment of logic, and neither do they represent any interesting NLP task. This can be simply seen as booleanized and very restricted version of  Chain of Thought reasoning, which works for a few steps and then does not work for deeper reasoning tasks, which is what the authors conclude.

**Questions:**

-  Have you tried boolean expressions which may have more than one proof?
- In what way can your results aid designing new methods --- prompts, training or theory--- for LLMs?

---

### Official Review · Reviewer_PEvF · 2023-10-31

**Soundness:** 2 fair
**Presentation:** 3 good
**Contribution:** 2 fair
**Rating:** 3
**Confidence:** 3

**Summary:**

This paper shows that large language models (a) fail to generalize the compositional structure and (b) often fail not only in calculating formulas but in grasping the input structure itself.
The aim is to ascertain whether a language model can employ basic elements to construct a logical framework.

**Strengths:**

The paper is asking a very interesting and fundamental question about LLMs.

The methods used are basically sound.

**Weaknesses:**

I am confused about the underlying tasks. The paper addresses 2 totally different tasks:

1.  What can an LLM represent in relation to logic relations

2. What proof properties exist for an LLM.

The main body of the paper lacks a proper theory for the language model and the notion of compositional reasoning in models.

The boolean representation you use should be defined at the beginning, and not in sec. 4.1 (Experiments).
It seems like this is a more general representation of formulae, like an abstract syntax tree (AST). Here, the tree represents the abstract syntactic structure of text  written in a formal language. Each node of the tree denotes a construct occurring in the text.

I would suspect that using the AST is closest to what an LLM creates. Is this the intention? If so, then this compromises any proof methods.

The claim "The advantage of representing boolean algebra as a tree lies in dividing the boolean algebra set according to its complexity, represented by depth." depends totally on the representation. The proposed representation makes no guarantees of depth-minimality. A lot of work has been done on "optimal" boolean reprsentations, e.g., Binary Decision Diagrams (BDD).

Thee is no precise definition of compositional reasoning in models. Is there a target definition of compositional reasoning?
The claim on p. 5 "This phenomenon demonstrates the compositionality of boolean algebra." makes no sense without a  precise definition of compositional reasoning.






Problems also arise in task (2), where you "examine the failure points of the model’s proofs".

There is no definition of "proof" or proof method. Hence we cannot understand what you mean by task 2.

There are several recent papers on LLM and Proofs, e.g.,

Xiong, J., Shen, J., Yuan, Y., Wang, H., Yin, Y., Liu, Z., ... & Liu, Q. (2023). TRIGO: Benchmarking Formal Mathematical Proof Reduction for Generative Language Models. arXiv preprint arXiv:2310.10180.

How does your work compare to these?

The experiments on task 2 (proofs) do not address this task. Please see the above cited paper for a more rigorous method.

**Questions:**

1. Would the approach fare better if you used as a representation:

     (a) Disjunctive Normal Form (DNF)

    (b) Binary Decision Diagrams (BDD)

2.  Do you intend to look at the learning ability of de Morgan's logic rewrite rules, or more complex theorem proving methods?

---

### Official Review · Reviewer_iCBt · 2023-10-31

**Soundness:** 4 excellent
**Presentation:** 4 excellent
**Contribution:** 2 fair
**Rating:** 6
**Confidence:** 4

**Summary:**

This paper proposes a way to evaluate compositionality in Language Models.
The main idea is to test the ability to induce the value of a complex semi-linguistic structure by using the value of the components. The target of the analysis is boolean expressions. LMs are exposed to expressions up to a given level for learning. At inference time, these LMs are queried with expressions at a major level of complexity. This is used to test whether or not LMs learn the compositional behavior.

**Strengths:**

- The paper clearly demonstrates that LMs are not able to generalize

**Weaknesses:**

- This is only a negative paper: given the finding, there is no solution.

- The paper does not cover an important related part: Memorization

**Questions:**

- Is there a way to solve the limitation that you discovered?

- How do you relate your work with the idea that large part of these LLMs are memories: see: Data Contamination https://arxiv.org/abs/2203.08242 and PreCog: https://arxiv.org/abs/2305.04673

- How pre-training of GPT2 and T5-base can be helpful?

---

### Official Review · Reviewer_kQgJ · 2023-11-01

**Soundness:** 3 good
**Presentation:** 2 fair
**Contribution:** 1 poor
**Rating:** 3
**Confidence:** 4

**Summary:**

This paper evaluates small language models’ (Roberta, T5, GPT-2) depth generalization ability on Boolean expressions when fine-tuned with shorter depth Boolean expressions. In the main experiments, the training set consists of expressions with depth-1 and 2 circuits, whereas the test evaluations contain Boolean expressions with depth 3 and 4. They find that this model fails to generalize longer depth expressions in both the classification and the generation version of the task. They also have initial experiments with prompting LLMs in the Appendix F4.

**Strengths:**

1) The paper demonstrates generalization failure of small language models.
2) The paper is written clearly.

**Weaknesses:**

1) The paper does not contribute to the existing literature on compositional generalization in neural models and language models (see my questions on this and Furrer’s and Kim’s surveys for references); and does not evaluate the advanced techniques developed, for example Drozdov prompting work. The paper can also benefit from using the other length generalization papers. I am attaching some representatives that could have been included:

- Drozdov et al. “Compositional Semantic Parsing with Large Language Models”
- Liu et al. “Learning algebraic recombination for compositional generalization”.
- Anil et al., “Exploring length generalization in large language models”,
- Newman et al., “The EOS decision and length extrapolation”.

References for some other compositional generalization work that I think is not successful at recursive generalization:

- Russin, “Compositional generalization by factorizing alignment and translation”.
- Conklin, “Meta-learning to compositionally generalize”.
- Csordas, “The devil is in the detail: Simple tricks improve systematic generalization of transformers”.
- Akyurek and Andreas, “LexSym: Compositionality as Lexical Symmetry”.


2) The paper refers to the general category of language models, and the paper states in their contribution *“This demonstrates that language models struggle to understand and generate compositional structure, which implies that the recent achievements in reasoning are not a result of language models' systematical and structural understanding of tasks.”*.

Yet, the paper’s main experiments deal with small and relatively old models on a single fine-tuning setting. Moreover, none of the paper that the phrase “recent achievements in reasoning”  refer use Roberta/T5/GPT-2, they instead use Palm and GPT-3. So, I cannot see how the claims you make derived from these results alone.

So, I believe important experiments are needed to validate these claims and strengthen the contribution of the paper.

    - How does the increasing size of an LM affect these results? Do we see better generalization with increasing size?
    - What about mid-size, newer models trained with updated datasets  (LLama, Pythia series).
    - What about LLMs with prompting (you have only baseline results in Appendix F4, I suggest moving them to the main body. There are many methods on prompting nowadays gives significantly better results than few-shot prompting, see Drozdov et al., 2023).
    - What about LLMs with fine-tuning (GPT had an API for FT).


3) I think Table1, 2 and 8 could be summarized in a single plot which makes the presentation nice. My initial thoughts:
- X axis is `n` where n is maximum training depth
- Y axis include three connected dots per model; first dot is the accuracy at `depth<=n` test examples, second that is accuracy at `depth=n+1` test examples, and the third dot is the accuracy at `depth=n+2`
- Color can specify the different models
- Style can specify the pre-trained vs non-pretrained

You could play with these settings, but the essence is that you can compress many tables nicely into plots that are more informative about the generalization gap w.r.t function of `n`, model and pretraining.

**Questions:**

1) In the end of the intro, Contribution-1 includes many claims/contributions and they switch between *“Transformers …”* and *“Language models …”*. What is the reason for this switch? Is this paper about the limitations of Transformers trained from scratch or the limitations of LMs?

2) Contribution-2 includes an argumentation *“This prevents the model from forming a generalizable representation of compositional inputs.”*. First, I suggest moving arguments to other sections. Second, I am not sure what *“generalizable representation of compositional inputs”* refers to and where did you show something about the representations of the model?

3) Overall, can you make your contribution statement more concise and include only the concrete contributions of the paper?

4) How does this paper’s contribution differ from previous surveys on LMs’ compositional generalization ability?

    a) Furrer et al., “Compositional Generalization in Semantic Parsing: Pre-training vs. Specialized Architectures”,
    b) Emin Orhan, “Compositional generalization in semantic parsing with pretrained transformers”.
    c) Najoung Kim,  “Uncontrolled Lexical Exposure Leads to Overestimation of Compositional Generalization in Pretrained Models”.

Is the difference evaluating models on the Boolean dataset used in the paper? If the answer is yes, how it fundamentally differs from COGS (Kim and Linzen, 2020), especially from COGS’s recursive generalization?

5) How does this paper’s dataset and experiments differ from the following paper on generalization of Transformers on propositional logic: Schlegel, et al. *"Can Transformers Reason in Fragments of Natural Language?."*? Is the difference Boolean algebra vs propositional logic?

6) Why do you have a subsection as *"Propositional logic and First-order logic"* in the Background. Do you evalate on a propositional Logic dataset that I missed?

**Summary of the Review**

Overall, I think this paper does not add to the contributions of the following papers on LMs and compositionality (Furrer et al.; Orhan; Najoung). I delineated more related work that needs to be distinguished in the review. The paper claims about LMs without running state-of-the art LMs and techniques. Therefore, I suggest authors conduct the suggested experiments and revise the paper accordingly.